# Anxious Apocalypse: Transmedia Science Fiction in Japan's 1960s

**Brian White**

Department of Japanese, Kalamazoo College, Kalamazoo, MI 49001, USA; brian.white@kzoo.edu

**Abstract:** Science fiction (SF) developed as a self-identified genre in Japan in the 1950s and quickly underwent a boom in the 1960s. Throughout this period, SF literature, film, and television were tightly intertwined industries, sharing production personnel, textual tropes, and audiences. As these industries entered global circulation with the hope of finding recognition and success in the international SF community, however, they encountered the contradictions of the Cold War liberal cultural system under the US nuclear umbrella. Awareness of the discursive marginalization of Japanese SF in the Euro-American dominated global SF scene manifested in Japanese texts in the twin tropes of apocalypse and anxiety surrounding embodiment. Through a close reading of two SF films—*The X from Outer Space* (*Uchū daikaijū Girara*, 1967) and *Genocide* (*Konchū daisensō*, 1968), both directed by Nihonmatsu Kazui for Shochiku Studios—and Komatsu Sakyō's 1964 SF disaster novel *Virus: The Day of Resurrection* (*Fukkatsu no hi*), I argue that, largely excluded from discursive belonging in the global community of SF producers and consumers, Japanese authors and directors responded with texts that wiped away the contemporary status quo in spectacular apocalypses, eschatological breaks that would allow a utopian global order, as imagined by Japanese SF, to take hold.

**Keywords:** 1960s; Apocalypse; Cold War; discourse; embodiment; SF; science fiction

## 1. Introduction

When discussing science fiction (SF) in Japan in the 1960s, it is difficult to contain one's attention to any single medium. Science fiction literature, film, and television were tightly interconnected industries, drawing on similar personnel, textual tropes, fan bases, and discourse communities. With the release of *Godzilla* (*Gojira*) in 1954, the influential but short-lived run of the SF fanzine *Seiun* in the same year, and the founding of the two longest-running fan and professional SF periodicals *Uchūjin* and *SF Magazine* in 1957 and 1959, respectively, the genre took identifiable shape in Japan in relatively short order in the 1950s as a transmedia assemblage that visualized a set of ideological orientations toward the future, as well as toward the present that would bring that future about. As the 1960s progressed, this transmediality would only grow more extensive, with manga and anime joining film and television to become a prominent part of the SF visual landscape. SF literature and the media industries in Japan, in other words, have always fed into one-another.

This close relationship between media was frequently a source of productive cooperation. For instance, the first professional SF literary submission contest, organized by *SF Magazine* in 1961, offered as its grand prize not only a guarantee of publication within that magazine, but also the right to work with staff from Tōhō Studios to adapt the winning work into a film. More generally, individual SF creators frequently crossed media boundaries in their work. Ishinomori Shōtarō is a prominent example, finding success as both a manga writer and as a writer for *tokusatsu* special effects TV series like the wildly successful *Kamen Rider*. He was not unique, however, and Komatsu Sakyō, Hoshi

Shin'ichi, Tezuka Osamu, and many other prominent SF authors were also regular contributors to the small screen.

For film and TV studios, this arrangement provided personnel, original film treatments, and works for adaptation to fill a large studio demand for science fiction blockbusters that could be sold overseas in order to help the Japanese film industry recover from the serious financial difficulties that had been plaguing it throughout the decade. For authors and publishers, meanwhile, it provided more widespread attention thanks to the significant reach of film and television, as well as producing concrete visualizations of the SF worldview that could serve to further inspire authors toward greater literary production and fans to greater transmedia consumption. In the numerous columns on science fiction film, television, and radio within SF literary magazines within the decade, we can see a genre that was always understood to exist in the overlap of the written word and visual expression, and indeed to heighten their interplay.

Yet if this were largely true in the domestic sphere of SF production and consumption, as Japanese science fiction media began circulating internationally, anxieties among its literary coterie quickly began to come to the fore. By the end of the decade, prominent figures in the SF literary community complained that the flashy special effects and sophisticated miniatures work that characterized SF film threatened to doom the genre as a whole to an image of shallow, un-thoughtful popcorn entertainment churned out by film studios that, far from sharing their ideological commitments to a brighter future for Japanese society, were simply chasing quick profits with films that paled in artistic comparison to their foreign counterparts.[1] Across the Japanese SF literary world, there was a sense that the genre was failing to achieve the implicit goals of finding international (especially Euro-American) recognition and being accepted into the fold of the international community of SF. In other words, we could say that a major goal for both SF literature and image media was international success, but that the specific metrics by which that success was to be measured differed. For the film and TV industries, it was the financial success that came from widespread distribution in foreign markets. For literature, it was aesthetic recognition within the global science fiction community and inclusion in the SF canon alongside names like Heinlein, Asimov, and Lem.

Attending to the formal and narrative characteristics of a few representative examples of SF film and literature from the decade, however, reveals that there is not as much ideological daylight between the two industries' goals as the above critiques would lead us to believe. These texts shared commitments to American-style liberal multiculturalism, modern rationality, and humanist universalism, but it was precisely their attempts to engage with the US markets that posed a manifest problem for each of these ideological positions. While each of the texts we will examine here held out some level of hope for humanity, their optimism was shrouded in varying levels of ambivalence that demonstrate the ethical and intellectual impasse Japanese SF producers found themselves facing in the 1960s.

Instead of the differing aesthetic and commercial ambitions of the two industries of SF film and SF literature, then, it makes more sense to consider Japanese SF as a transmedia industry that entered global circulation as a body of national texts. Regardless of the specific aims of intercourse with the global SF community held by authors and film studios, the Cold War moment of the 1960s meant that the geopolitical conflict between the US and USSR inflected the ideological position of SF creators in Japan, a US ally. Even as the Japanese SF discourse community saw itself as a potential bridge that could span the Iron Curtain thanks to its geographical proximity to the USSR and political alliance with the United States, the utopian futures so frequently illustrated in the pages of SF periodicals of the time overwhelmingly saw those utopias being built on advanced consumer technologies that enabled the apotheosis of liberal capitalist lifestyles centered on the nuclear family as the hegemonic social unit. Japanese SF texts aligned themselves ideologically with the US markets into which they were trying to find entry. The realities of

US hegemony, however, meant that these creators could not find acceptance abroad as true equals.

Below, I present a comparative analysis of two science fiction films—*The X from Outer Space* (*Uchū daikaijū Girara*, 1967) and *Genocide* (*Konchū daisensō*, 1968), both directed by Nihonmatsu Kazui—as well as Komatsu Sakyō's 1964 disaster novel *Virus: The Day of Resurrection* (*Fukkatsu no hi*) in order to demonstrate the ways each text understood its Cold War moment. In spite of the fact that the films largely embody the trends criticized by the SF literary mainstream while Komatsu's novel epitomized their prescriptions for the genre, the three texts end up at many of the same political and ideological conclusions. Visualized and narrated in the SF cultural production of the '60s was a desire for a radically future moment disconnected from the lived present of its creators, which is taken to be increasingly hopeless. The political present in these texts can be resolved only through its own effacement and destruction, an apocalyptic escape hatch from a moment in which Japanese SF producers were caught in the contradictions of the liberal world view of the American camp of the Cold War. These texts clung to a conservative racial and sexual politics surrounding human embodiment, however, which served to reinscribe the conditions of the very marginalization against which they struggled. In order to illustrate my argument, let me turn first to the two SF films.

## 2. Three Apocalypses

Despite being produced by the same studio and sharing a director, cinematographer, and some cast members, *The X from Outer Space* and *Genocide* are markedly different in tone and themes. Two of 15 total films produced by Shochiku to receive government funding under a program intended to buoy a flagging film industry by encouraging films for export, they received a total of 251.3 million yen according to Tanikawa Takeshi. (Tanikawa 2016) A third SF horror film produced by Shochiku during this period, *Goke, Body Snatcher from Hell* (*Kyūketsuki gokemidoro*, 1968, dir. Sato Hajime) was submitted for consideration for program funding but ultimately denied. As a studio, Shochiku was much better known as a producer of family dramas like those of director Ozu Yasujirō, so its presence among the recipients of funding for SF horror and monster films comes as something of a surprise. While the studio was far from the most prolific producer of SF films that received governmental support, the fact that it produced these three films with the aim of securing funds corroborates Tanikawa's conclusion that SF blockbusters were seen as one way to access potentially lucrative international markets. Together with one other film—*The Living Skeleton* (*Kyūketsu dokuro sen*, 1968, Matsuno Hiroshi dir.)—these would be the only SF films Shochiku would produce, suggesting a brief moment from 1966–1968 when the genre seemed attractive and marketable enough to the studio to cause it to consider pivoting away from the domestic dramas that had been its mainstay.

*The X from Outer Space* is a fairly straightforward tale of the Earthly invasion of an alien monster, Girara. An international science team, sent to Mars to investigate the disappearance of other research teams and sightings of UFOs, returns to Earth carrying an alien spore that mutates into the towering, destructive Girara and rampages throughout the Kanto and Tohoku regions of Japan. In the background of this narrative, a parallel story of a love triangle among the crew between captain Sano, Lisa, and Michiko plays out, with the tense prospect of interracial romance between Sano and Lisa eventually resolving to the more conservative pairing of Sano and Michiko, with Lisa confiding in the white project lead Dr. Berman that she has resigned herself to the outcome. Throughout, the film is conspicuously upbeat, backed by an up-tempo score resembling samba music and with Girara's destruction of large swaths of Japan getting fairly minimal screen time. The will-they-won't-they story of the Lisa-Sano-Michiko love triangle seems just as narratively weighty, if not more so, than the 12-story tall, solar-powered lizard chicken laying waste to Tokyo. In short, in addition to hewing somewhat closer to Shochiku's bread-and-butter topic of romantic drama, *The X from Outer Space* also embodies the racially and sexually conservative trappings of much of mainstream SF in Japan. Scientific reason

triumphs, and the outside threats (alien and woman alike) are contained. Notable is the parallel drawn between Girara and Lisa at film's end. Girara is not destroyed, but rather shrunk back down to its dormant spore state, sealed into a container, and then blasted out to the far reaches of space aboard a rocket. Similarly, no climactic, violent confrontation ever occurs between Lisa, Michiko, and Sano. Instead, Lisa simply walks off morosely with Dr. Berman—possibly returning with him to wherever he flew in from in the opening scene of the film—while Michiko and Sano stand gazing toward Mt. Fuji in a softly nationalistic romance. There is no explicit strife; Lisa simply goes away.

*Genocide*, released a year later, is in many ways the polar opposite of *X*, and it might therefore be seen as emblematic of more "mature" SF in the vein of Komatsu's apocalyptic tales of the fate of humankind, which we will explore below. An American B-52 bomber carrying a hydrogen bomb crashes near the Anan archipelago, islands whose tropical scenery and American military presence would surely evoke Okinawa to contemporary viewers. Two apocalyptic conspiracies unfold from this crash. On the one hand, the jingoistic and secretive American Colonel Gordon searches desperately to recover the lost atomic bomb before news gets out of its disappearance and causes America to lose face in the Cold War battle of reputation. On the other, the conniving femme fatale Annabelle, having developed a nihilistic hatred of all humankind after her experiences as a prisoner in Auschwitz, is creating new breeds of venomous insects, a cloud of which were responsible for bringing down the B-52. Her hyper-advanced insects have somehow become psychically active, seeking to destroy humanity rather than allow nuclear proliferation and war to end all life on Earth. At the center of these two plots is Jōji, an insect collector who has been assisting the Tokyo-based entomologist Dr. Nagumo as well as (unwittingly) Annabelle, with whom he is involved in an extramarital affair. Nagumo becomes the hinge connecting the American military to the scientific mystery of the insects. The film ends with a repentant Jōji martyring himself to save his pregnant wife Yukari from the insects that were set loose through the death of Annabelle. Yukari flees the island on a small boat just before Colonel Gordon remotely detonates the bomb as a last resort measure to keep it out of enemy hands. Yukari, alone on her tiny motorboat, appears to be destined to be the sole surviving human as nuclear conflict and ecological judgment day both threaten the genocide of all human life.

Whereas *The X from Outer Space* was a largely optimistic cinematic romp, *Genocide* takes a darker turn, positing Cold War polarization and ecological destruction as spelling the ultimate doom of the human race. Uniting both is a charged theme of interracial romance (Lisa and Sano, Annabelle and Jōji), but I would argue that, more than a fear of miscegenation, it is the Cold War system itself that makes the difference between utopia and dystopia in these films. Any hint of real-world political divisions is entirely absent from *X*, whereas it is precisely American pigheadedness with regard to their constant paranoia about "Red agents" on Anan that leads to nuclear annihilation. The Cold War is not an overdetermining factor in Shochiku's SF thematics, however. Rather, it is one of a number of different symbols of human division more generally, most notably including the war in Vietnam (journalistic photographs of which are used briefly in *Genocide*). Like much SF of the decade, Shochiku's films present a stark choice to the human race: unite or die.

These same themes were highlighted in Komatsu Sakyo's 1964 novel *Virus: The Day of Resurrection*. Komatsu's book narrates the end of the world as brought about by a secret, bioengineered virus, and serves as a meditation on the incompatibility of a Cold War factionalist mindset with the reality of the natural world. Throughout, inclusive liberal rationalism is emphasized as the most ethical mode of being for the contemporary moment. The ethics of intellectualism and scientism, ethics that were being actively promoted in the pages of the professional periodical *SF Magazine* at the time, are epitomized in *Virus*, becoming the means by which a new and better world can be built upon the ruins of the old. The text serves as an ambivalently cautionary tale in which the values of SF that were being articulated in the pages of publisher Hayakawa Shobō's monthly magazine might

not be enough to prevent apocalypse but could at least ensure the continuation of the human race in spite of it. Democratic, scientific intellectualism, when applied as a model for communal human existence as a whole, points human beings on an upward developmental trajectory, securing the promise of an enlightened and prosperous future even after society as we know it is destroyed.

The text's main cast is an international group of scientists stationed in Antarctica. They are reminiscent of the pointedly multicultural cast of *Star Trek* (1966–1969), and like Gene Roddenberry's television series the group is used as a model of a utopian future. *Virus* concerns itself with the promises and dangers of connection and isolation in a Cold War world. It is explicitly the isolationist secrecy of the world's militaries—specifically those of the US, Britain, and the USSR—that is responsible for allowing the virus to spread; military reticence to reveal their secret bioweaponry research hampers the international medical response to the epidemic, and by the time civilian doctors discover what is happening, it is already too late. Military-industrial information hoarding is depicted as an existential threat to the world.

By contrast, the civilian scientist characters are democratic and open in sharing and aggregating the knowledge they produce. Upon realizing that they are the sole remaining members of the human race, they quickly and efficiently organize a small-scale society among the Antarctic research bases, using logic and scientific rationality to delegate labor, distribute resources, maintain social order, organize research outings, and systematize reproduction. Old national identities are erased in favor of a post-national, post-racial model of social equality and self-sacrifice in the name of the greater good. When a fluke of luck causes an automated nuclear response system cascade to eradicate the super-virus, the scientists eventually make their way outside the Antarctic to repopulate the globe in line with this new, enlightened scientific social order.

Komatsu's worldview is one that emphasizes a fundamental contiguity between all people, one that is intensified under the material conditions of modernity. A scene from the beginning of the novel represents this global connectivity through air travel. Komatsu visualizes railways and air traffic routes as circulatory vectors for disease transmission. At the close of the first chapter, the biological superweapon has been released into the world—fittingly, because the spy plane carrying it has crashed in the Italian Alps—when the focus of the narration drifts away from the site of the wreckage, as though the text itself is the virus, being carried up and dispersed by the jet stream. It spends a short time describing the rivers, mountains, and other natural boundaries that divide Europe before shifting to a similar description of rail lines and airports:

> The railway that passes by Italy's northern entry point of Torino ran west by way of Milan, passing through Venice, Trieste, Beograd, and Sofia on the way to Istanbul, gateway to Asia, then turning southwest, past Genoa and the eastern coast of Italy on the way to Rome and Napoli. To the east, it ran through Lyon and Dijon to Paris, heading into the very heart of Europe. From Milan, there was also a line that ran through the famous Simplon Tunnel to arrive at Lausanne and Geneva in Switzerland. All of middle and eastern Europe was bound together in a net of railways. In the great cities of Europe—Rome, Paris, Geneva—there were international airports where streams of people flew down from the sky and back up into it, flowing like great rivers…(Komatsu 2012, pp. 48–49)

Komatsu's description deconstructs two types of boundaries: political and epistemological. On the one hand, the flows of international travel as embodied by the rail and air routes enumerated here effaces the political boundaries between European countries—boundaries which demarcated the Cold War lines of political alliance that will be one of the main drivers of conflict in the rest of the novel and which must be overcome in order to ensure the survival of the human race. Even as the passage lists off the names of cities and states within Europe, these names flow by as though seen from a train window, scenery that zips past without at all impeding the reader's smooth travel. On the other, the

naturalistic description of air travel, in which flows of people are compared to "great rivers," stresses the contiguity between the traditionally binary categories of natural and artificial. The people of Europe here are connected on a fundamental level not only with one another but also with the natural world. Their migrations are refigured as features of the natural landscape that, while sometimes serving as the basis for political boundaries, just as frequently cross those boundaries.

Given the apocalyptic implications of the virus that the reader has already gleaned, this seems less like a sunny meditation on the border-dissolving potentials of international travel and global connectivity, and more like an epidemiological diagnosis of the virus's spread. Thus, we can see that, for Komatsu, it is not the fact of global connection itself that holds the emancipatory potential, but the ability and willingness of human beings to recognize that potential and pursue it. Put another way, humans must recognize and embrace the circulatory pathways of culture and society, lest these arteries become infected with something more sinister and destructive.

The cure for these ills is none other than the "intellect" (chisei) that the professional science fiction literary venue *SF Magazine* promoted among its readers. The novel's faith in scientific rationality and liberal ethics as organizing principles for human society is a consistent theme throughout the book, and Komatsu depicts scientists who are working to freely disseminate knowledge as unambiguously good. Scientific knowledge and its proliferation are taken as the basis on which a just society can be built, overcoming all obstacles of national or racial division. *Virus*'s optimistic ending paints a hopeful picture in which the world is able to transcend the national divisions and militaristic secrecy of the Cold War to create a society built on a fundamental basis of intellect.

Stylistically, *Virus* is written in the same empirical realist mode as the majority of the fiction being published in both professional and amateur science fiction magazines at the time. A limited-omniscient third-person narrator aligns the reader's experience of the virus's spread and the global reaction to it with the perspectives of an array of characters that are caught in the middle of the crisis. The reader is given access to the internal thoughts of the character with whom they are aligned, allowing them to see how, for example, the doctors encountering the virus arrive through ratiocination at an understanding of its epidemiological properties. The scientists and doctors who serve as the main objects of reader sympathy are unfailingly rational in their thought processes, using evidence-based reasoning to deduce what must be done. This aligns with the general narrative arc of many of the stories that appeared in *SF Magazine* at the time, in which the protagonist is able to uncover the truth of a mysterious phenomenon only after careful investigation. A narrative structure of discovery combines with an ethical stance of humanism in *Virus* to produce a resolution in which the protagonists uncover the ultimate truth that, more than the deadly virus, it was the fact of humanity divided against itself that was the largest obstacle to human thriving across the globe.

## 3. The Intolerable, Inescapable Present

In all three texts, the future of the contemporary global political climate of the 1960s was unimaginable. For *Genocide* and *Virus*, the present is wiped away in nuclear, ecological, and viral apocalypse, clearing the way for the utopian social order to take hold. Interestingly, however, these twin apocalypses, reacting to the same intolerable moment, end us up at contrasting conclusions. In Komatsu's novel, the viral end of the world returns it to a pre-political (or advances it to a post-political) moment wherein geopolitical distinctions between states are wiped off the map and human individuals can grasp their own interconnection with each other and with the natural world, a realization that is assumed to point them toward altruism and obviate material needs and inequality. Conversely, Nihonmatsu's film seems to suggest it is precisely the multiply ambivalent contact zone of Anan—neither American nor Japanese territory, contested at the boundary of Cold War ideologies, threatening the monoethnic body politic with dangerous multi-racial extramarital affairs—that creates the conditions of apocalypse. Nuclear annihilation, in this



sense, eliminates this space of ambiguity and reasserts the geopolitical separation of nation-states. While *Genocide* ends on a much more pessimistic note than *Virus,* it holds open the possibility of a new beginning for the human race through Yukari, pregnant with the child she conceived with Jōji and perhaps ready to stand in for a more conservative model of the nuclear family in line with a nationalist conception of biopolitics. Both texts threaten the end of the world as the result of Cold War divisions, but their visions for the post-apocalypse seem to diverge in whether or not the nation-state structure remains as the organizing principle of human society.

In *The X from Outer Space*, meanwhile, the present is simply willed into a more amenable future configuration, with national and political difference almost (though not quite entirely) swept out of sight. Our scientist protagonists are an international team—all speaking perfectly fluent Japanese—and while it is Japanese territory that Girara threatens, it is for the sake of the world, and with the cooperation of the United States at the very least, that the scientists defeat it. We are even told in a brief aside that the United States "doesn't want to drop another nuclear bomb on Japan" to stop the creature, presumably feeling remorseful for having done so in 1945. Japan simply begins the film as an already accepted equal among the space-faring superpowers, complete with its own lunar base. How the tensions that had been on display only a few years before during the protests against Anpo (the US-Japan Joint Security Treaty, against whose renewal protesters nationwide had mobilized) had been resolved in order to deliver us to this future moment of harmonious cooperation is left unexplained. Whereas *Virus* and *Genocide* had only been able to imagine the future after a fundamental break with the present, *The X from Outer Space* simply whisks political conflict out of sight.

This political sleight-of-hand would be repeated in the 1970 International Exposition in Osaka (Expo '70), where sleek prognostications of a comfortable consumerist future were everywhere to be found, while the edges and points of political critique were rhetorically sanded off of the "Wall of Contradictions". (Gardner 2011) A brighter future was possible at the Expo, it seemed, only insofar as it did not trouble the present status quo. At the International Science Fiction Symposium organized by members of the Japanese SF community and held concurrently to the Expo, too, topics of conversation remained largely de-politicized, with discussions instead focusing on, for instance, industrial pollution in a more abstract register. (Tatsumi and Kurabu 2015) Why did science fiction, a genre explicitly concerned with imagining the future of society, find it so difficult to articulate a path toward that future that accounted for the political and material conditions of the present? Why did SF have so much trouble fulfilling Abe Kōbō's desire, articulated in 1966, for it to be "unnameable" and thus capable of a materialist politics and meaningful social critique? (Abe 2002) A closer look at the genre's visual aesthetics in the 1960s will help us explore these questions.

## 4. Anxious Bodies

As noted above, both of the films under consideration in this article treat sexuality very conservatively, and this stance extends to the human body itself. We have already observed, for instance, the racially and sexually conservative ending of *The X from Outer Space*, but even in the more soberly critical *Genocide*, deviant or disruptive bodies are treated with suspicion and violent disavowal. Beyond femme fatale Annabelle—who threatens the possibility of extramarital miscegenation with the protagonist Jōji in addition to her role as metonym for science run amok—there is the sole survivor of the plane crash that starts the film, a black GI named Charlie. One of the first victims of the psychic wasps bred by Annabelle, Charlie exhibits psychosis due to the insects' venom. He eventually escapes the hospital where he is being treated and stumbles about the area wildly firing his pistol. Indiscriminately violent, insane, and sexually predatory (he attempts to rape the nurse who had been treating him), Charlie's very existence threatens the island, and he is eventually shot and killed.

Elsewhere in the film deformations of the human body are only acceptable under strictly controlled circumstances. Dr. Nagumo, in an effort to understand the mysterious wasps, concocts a plan to allow himself to be stung in order to test the psychotropic effects of the venom. He has Jōji and Yukari tie him to a chair so that control can be maintained over him while he is under the effects of the venom, unlike in the case of Charlie. Also unlike Charlie, Nagumo's psychosis is short-lived and far less violent: rather than exhibit any aggressive or anti-social behaviors, he simply seems to enter into a trance. With experimental safeguards in place, in other words, Nagumo's body never becomes uncontrollable, nor does it threaten the bodies of those around him.

Beyond the human bodies of characters in SF film, too, even alien embodiment tends to remain comfortingly familiar. *Tokusatsu* SF films were widely known for their extensive use of miniature sets and monster suits in order to visualize gigantic alien creatures laying waste to the urban centers of Japan, and the particular sense of uncanniness that accompanies miniatures-based special effects photography in films like *The X from Outer Space* deserves closer scrutiny here. As we witness Girara stomping around Japan and eating power plants, we are meant to understand that it is hundreds of feet tall thanks to the inclusion of scale models of buildings and natural landscapes. Even still, however, the eminently humanoid structure of Girara's body, the curious ways its skin folds and deforms as it moves, and the sense of weight imparted (or rather, not imparted) by its movements reminds us that there is a human inside the suit. What is more, the physicality of the model buildings as they are toppled and thrown about *feels* miniature, toy-like. The buildings bounce across the ground, for instance, and scenery elements such as trees remain curiously intact when Girara's lightweight, pliant foot comes down on them. While Paul Virilio (Virilio 2009) decries the ballistic optics he sees as characteristic of cinema because of their *inhuman* nature that rationalizes and accelerates the lifeworld to the point of uninhabitability for human beings, what *X* demonstrates to us is conversely the persistence of human scale. At the risk of merely repeating a truism, science fiction was of course a genre that was deeply concerned with human society and human embodiment, in spite of its narrative attention to otherworldly beings.

Japanese SF in the 1960s, then, was ultimately unable to overcome embodiment as a source of anxiety. We do not forget Girara's human core, and it is perhaps for this reason that it is simply encased in foam and shot into space rather than being destroyed more thoroughly. The human body was to be protected from destruction or alteration—shielded from any serious deconstruction or critique as a site of identification and difference. At the same time it was precisely the marked embodiment of Japanese SF creators that seemed to foreclose the possibility of Japanese science fiction's circulation within the hegemonically Euro-American sphere of global SF. National, racial, ethnic, and linguistic difference haunted Japanese SF as it circulated globally, with the multiculturally liberal future it envisioned always just out of reach beyond the walls of these differences. The SF community in Japan hesitated to discard the human body and the meanings attached to it, but embodied categories of difference always kept them at a remove from their counterparts across the Pacific. The unequal relations of power that could accrete to that separation was a continuing source of frustration and anxiety for Japanese SF producers and fans.

We see this anxiety play out in a travel report made by the editor of the prominent fanzine *Uchūjin*, Shibano Takumi. In 1968, the year following the defeat of Shibano's bid for a pan-Pacific SF fan convention (Pan-Pacificon) as the next installment of the global SF convention Worldcon, he undertook a 50-day tour through the United States, meeting with prominent figures in the American SF fan scene, as well as SF authors. There, in collaboration with "big-name fans"[2] John and Bjo Trimble, Forrest J. Ackerman, and others, he established the Trans-oceanic Fan Fund (TOFF), a financial entity through which Shibano hoped to sustain international interchange between fan communities.[3] One of the early major donations to TOFF came from a tongue-in-cheek fundraiser within the Japanese fan community from the "Chase Shibano Takumi Off to America Group" (Shibano

Takumi wo Amerika ni opparau kai), spearheaded by author Ōmiya Nobumitsu. The group raised 158,428 yen and 85 dollars, in addition to sundry non-currency donations of goods such as a hanging scroll donated by Mitsunami Yōko, one of the few female authors actively publishing SF in Japan during the decade. Shibano reports in his thank-you note to the donors that he donated almost all the money to TOFF. [4] Shibano's account of TOFF's founding and its activities would appear in a column almost every month for the better part of a year in *Uchūjin*. Next to photos of Shibano mingling with the major names of American fandom and photos of some of those same figures visiting Shibano's home in Japan, he wrote about his experience collaborating with his counterparts in American fandom.

A key figure in these columns is Roy Tackett. A longtime collaborator with Shibano, Tackett was an SF fan who had been stationed in Japan during the US Occupation, apparently making the former's acquaintance at that time. Shibano goes so far as to deem him "The Discoverer of Japanese Fandom" (Nihon fandamu no hakkensha), the colonialist resonances of which already hint at the implicit power disparity between US and Japanese science fiction fandoms. The idea for Shibano's Pan-Pacificon bid originally came out of correspondence between the two of them after Tackett had returned home to Albuquerque. (Shibano 1968, p. 15) Tackett acted as a contact point in the United States for Shibano, facilitating introductions between him and the other "big name fans" in the US. Yet even here, we can see evidence of an implicit hierarchy that places American fandom and its luminaries the Big Name Fans in the role of gatekeepers of success for Japanese fandom. Shibano quotes letters from Tackett and the other organizers of Pan-Pacificon and TOFF in his report. "We're not going to invite a Japanese fan because we want to discuss SF with him," they say. "We're bringing him over because he's someone we know and we'd like to meet in person. Our goals are friendship and amity," reads a quote from Tackett. An LA-area organizer, meanwhile, says, "As for you yourself, Takumi, while you're very well-known among American fandom, we still couldn't say that you're a BNF [Big Name Fan]." (Shibano 1968, p. 17) Shibano's activities within Japanese SF fandom over the preceding decade or more are apparently not enough to verify him as a Big Name Fan; for that, he would need to be authenticated by representatives of American fandom.

Shibano's account of his flight to America with his wife Sachiko under the auspices of TOFF, then, is marked by anxiety. He discusses his worries that his English won't be sufficient, or that he will miss the Americans sent to meet him at the airport (if indeed they even come: he wonders whether Americans share the Japanese custom of meeting people at the airport) and get lost. While he characterizes the trip as a whole positively and expresses overwhelming gratitude to the Japanese and American fans who funded TOFF, the first installment of his report nevertheless betrays a fear that he might somehow not live up to expectations placed on him by each group as a kind of delegate of SF fan activity.

When they land in Los Angeles, their first meeting with their American counterparts is a scene of dis-communication. The Americans' strongly accented pronunciation of the Shibanos' given names is rendered in katakana phonetic script, as is Shibano Takumi's halting responses to them in English as introductions are made. Shibano notes that he couldn't remember how to say, "Thank you for coming to meet us" in English, and so, "I just kept repeating the same greeting as though it were the only one I knew." (Shibano 1968, p. 25) While the scene is presented as humorous self-deprecation, it nevertheless suggests an ethnonationally-charged disparity between Shibano, envoy of Japanese fandom, and the white SF fans representing the United States. Even as Shibano undertakes his trip to bring together American and Japanese fan communities as equals, his report betrays the subtle ways in which the social dynamics that characterized SF production and consumption transnationally re-inscribed anxiety and a sense of inferiority on his part, despite his long years of experience working in Japanese SF production at the helm of *Uchūjin*.

## 5. Conclusions

Shibano's ambivalent experience in the United States is a poignant illustration of the contradictory situation in which Japanese SF and its producers found themselves as they tried to send their texts abroad and join the global SF community in the 1960s. Despite the promise of American liberalism that all could be equal participants in the global democratic order, Japanese SF authors and directors constantly found themselves marginalized relative to their white Euro-American counterparts. Consciousness of this problem frequently manifested itself within Japanese SF texts as an anxiety around the human body. Ethnonational categories of embodiment were willfully effaced in pointedly multicultural and multiethnic casts like those of *The X from Outer Space* and *Virus: The Day of Resurrection*, while nationalistically inflected sexual politics were violently reinscribed as in *Genocide*, resulting in a kind of optimistic pessimism that embodiment might be able to be temporarily ignored, but never totally overcome. Even if Lisa and Dr. Sano can work together as intellectual colleagues, their racially marked bodies push them apart and push the film (and Japanese SF more generally) toward the anxiously conservative stance toward human embodiment analyzed above.

At the same time, Japanese SF discourse was largely unable to imagine an outside to the Cold War political order in which the American-Soviet conflict overdetermined the boundaries of the SF community. For instance, almost entirely absent from Japanese SF in this decade is any serious engagement with Asia and Japan's relationship to it. Save for texts like *Mothra* (*Mosura*, 1961, dir. Honda Ishirō) that imagine Asia—and especially southeast Asia—as thoroughly premodern and thus not part of the technophilic future so central to the genre, Japanese SF was largely silent on the political possibilities of imagining the SF community to include Asian creators. Thomas Schnellbächer (2007) has discussed the Asian Pacific as a site of both a post-imperial return of the repressed as well as its spiritual resolution, while SF author Mitsuse Ryū (Mitsuse 1966) documented his travels to China in the summer of 1966 through a lens that saw the country's mythic past always superimposed over the present. Asia, in other words, was Orientalized as the past against which Japanese SF's futurity could be envisioned alongside that of Europe and the United States, in spite of the fact that these same tropes had historically been deployed as part and parcel of the marginalization that so frustrated Japanese SF creators.

If SF literature in Japan seemed to engage more actively and thoroughly with these contradictions than other genres of texts, it was precisely because of its entanglement with SF film and television. SF screen media—tied up as they were with global capitalist networks of production, circulation, consumption, and (especially) finance—pushed SF literature toward an international orientation. This was only heightened by the hegemonic generic tropes of science fiction, the Cold War liberalism of the "global village" that promised to overcome traditional barriers of language and nationality and perhaps even the nation-state itself. The combination of SF's genre form and its transmedia production apparatus encouraged global thinking, but when SF producers stepped onto the world stage, they would collide with the ways the Cold War liberal cultural system did not live up to its own ideals. Largely unable to find institutional or systemic purchase as equal contributors to the global sphere of SF, Japanese producers responded with texts that imagined the apocalyptic erasure of those systems, clearing the way for Japanese protagonists to step into leadership roles and re-shape global society along more utopian lines. The global perspective enabled and expanded by the transmedia connections between the Japanese SF literature and media industries prompted each to imagine a future free of the superpower hegemonies of the Cold War by which they were discursively marginalized.

**Funding:** This research received no external funding.

**Institutional Review Board Statement:** Not applicable.

**Informed Consent Statement:** Not applicable.

**Data Availability Statement:** Data sharing not applicable.

**Conflicts of Interest:** The author declares no conflict of interest.

## Notes

1. I have written elsewhere more extensively about this debate and its implications within the Japanese SF discourse community. See White, Brian. 2022. Mixed Media: Science Fiction and the Social Force of Genre. *PAJLS* 21: 75-85.
2. This is a common, informal title assigned to fans such as Shibano or Trimble who undertake a great deal of organizational labor to put together conventions and fan meetings, present awards, or organize international visits like Shibano's. It is often abbreviated as "BNF," and creates a soft hierarchy of celebrity within the fan community through the figure of the professional fan.
3. Such financial arrangements have precedence in TAFF (the Trans-Atlantic Fan Fund), which was started in 1952 and served as a model for TOFF.
4. This information and Shibano's note appear in a small fold-out note attached to the inside back cover of *Uchūjin*'s September, 1968 issue (issue no. 127).

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
