# Peer review of "Anxious Apocalypse: Transmedia Science Fiction in Japan’s 1960s"

_humanities, doi:10.3390/h12010015_

Round 1
Reviewer 1 Report
I read the article with great interest, and highly appreciate the effort to bring together close reading of texts, genre history, issues of transmediality, and actual experiences of the living actors at the time such as Shibano Takumi's visit to the US.
In your reading of the works, I was wondering if you could bring out more explicitly these two points:
- In your discussion of embodiment, the issues of gender and race play very important implied roles. Could you maybe add a sentence here or there that helps the reader see the significance of those markers and their implications both for the works discussed, and for the world-view (of Japanese SF or Japanese media/culture at large) that you present through them?
- In your description of how Japanese SF imagines itself within the Cold War world I was struck by the absence of Asia. Since it was both a constant presence in Japanese media at the time (think Korea, Vietnam, China, Indonesia, etc.), and it had figured so prominently in a previous iteration of that "Japan imagines itself as a global player" narrative that one could read in the Japanese Empire that ended in 1945, would it be possible to discuss what role Asia (especially E and SE Asia) played in the articulation of this "Cold War world" imagery deployed here. If it is entirely absent, the very fact of its absence would merit commentary too, in my opinion.
Congratulations on a strong article, and much success in the rest of your career!
Author Response
I appreciate your thoughtful responses to my article. I have added some language around lines 115-117 in the introduction to flag more clearly the importance of sexual and racial politics to the article's discussion of embodiment. I have also added a paragraph in the conclusion (lines 514-529) to briefly address the very apt point of Asia's notable absence from SF's discourses of futurity - an issue that deserves its own article.
Reviewer 2 Report
The article discusses three textual examples (two films, which are Girara and Konchu daisenso, and one novel, which is Komatsu Sakyo’s Fukkatsu no hi) from the 1960s Japan to argue that these texts show the author/director’s wish to destroy the current order and bring the utopian global order.
Thank you for the article. I very much enjoyed the comparative analysis of the apocalyptic texts from 1960s Japan. These texts receive much less scholarly attention from other contemporary texts while representing the emergent phase of the transmedial esu efu industry. The major weakness of the article is the ambiguity of the central argument. I’m not sure if I followed the author’s discussion especially when the topic shifts from the textual analysis to the real world outside the text and Shibata Takumi.
The ambiguity comes from many sentences, but especially from the argument such as: I argue that, largely excluded from discursive belonging in the global community of SF producers and consumers, Japanese authors and directors responded with texts that wiped away the contemporary status quo in spectacular apocalypses, eschatological breaks that would allow the utopian global order imagined by Japanese SF to take hold. (l.16-19)
After reading the essay, I’m not sure what the “contemporary status quo” is and what “the utopian global order” is exactly. Does the current order indicate the Cold War order? If so, how is the US-USSR divide (I thought it’s the Cold War divide) related to the US-Japan divide (or Japanese creators feeling excluded from the global SF fandom??)? Does the utopian global order mean people across national borders happily shake hands and forget racial or national differences? The two “orders” did not materialize in the author’s discussions as far as I understand.
In relation to the above point, a part difficult for me to follow was l.245-263 following the long block quote from Fukkatsu no hi. The quote was a description of cities and their networked railways in Europe seen from the sky. Then I wasn’t sure how the description was supposed to efface the political boundaries of the Cold War lines.
Another part I’m still struggling to understand is where it says “Japanese SF in the 1960s was ultimately unable to overcome embodiment as a source of anxiety (l. 390). The subject shifts from Girara looking at the human scale (typical of Tokusatsu) to Shibano Takumi visiting the US. The author needs to build more organic and logical connections to convince the reader with these examples.
Just for technical issues:
Does the author hit two spaces between sentences? I think it should be one space between sentences.
The femme fatale character’s name is spelled sometimes Anabelle and sometimes Annabelle. It just needs to be coherent.
Author Response
Thank you for your detailed comments. I am sorry to hear that my understanding of the "status quo" did not come across clearly. By this, I referred to Japan's position under the US nuclear umbrella in the Cold War and the prominence of ideologies of liberal multiculturalism, consumer capitalism, and humanist universalism that attended that political alliance and appeared frequently in Japanese SF texts of the decade. I have added and changed language throughout the article - especially in the three passages you cite explicitly - that I hope clarifies the relationship between this geopolitical situation and Japanese SF cultural production, as well as smooths the transition between my close reading of the film and literary texts and my account of Shibano's travels to the US. I hope the significance of that connection is made clearer there and in the conclusion, where I similarly revised my argumentation.
Reviewer 3 Report
Although the author discusses a couple of films and a novel very enthusiastically, he or she does not mention any of them in conclusion. The author should have closed the whole argument with a reminder that convinces us with the significance of these works. Without this strategy the author can't give us the sense of coming full circle and the author's digression into Takumi Shibano's TOFF trip to the US will gain no importance.Therefore, I strongly suggest that the author should revise the conclusion.
Author Response
Thank you for your comments. I have added language to the conclusion that ties the films and novel more closely to my discussion of the social dimension of SF production at the time. I hope this makes the relationship between the two halves of my argument clearer.